# Shared but Clean Household Toilets: What Makes This Possible? Evidence from Ghana and Kenya

**DOI:** 10.3390/ijerph19074271

**Published:** 2022-04-02

**Authors:** Prince Antwi-Agyei, Isaac Monney, Kwaku Amaning Adjei, Raphael Kweyu, Sheillah Simiyu

**Affiliations:** 1Regional Centre of Energy and Environmental Sustainability (RCEES), Civil and Environmental Engineering Department, School of Engineering, University of Energy and Natural Resources (UENR), P.O. Box 214, Sunyani BS0061, Ghana; 2Department of Environmental Health and Sanitation Education, Akenten Appiah-Menka University of Skills Training and Entrepreneurial Development, P.O. Box M40, Mampong 3JG3+PFQ, Ghana; monney.isaac@gmail.com; 3Department of Civil Engineering, Regional Water and Environmental Sanitation Centre (RWESCK), Kwame Nkrumah University of Science and Technology (KNUST), PMB, University Post Office, Kumasi AK448, Ghana; kaadjei.soe@knust.edu.gh; 4Department of Geography, Kenyatta University, P.O. Box 43844, Nairobi 00100, Kenya; rufmulaa@gmail.com; 5Urbanisation and Well-Being Unit, African Population & Health Research Center, P.O. Box 10787, Nairobi 00100, Kenya; ssimiyu@aphrc.org

**Keywords:** household shared sanitation, low income, cleanliness, basic sanitation, Ghana, Kenya

## Abstract

Shared sanitation facilities are not considered as basic sanitation owing to cleanliness and accessibility concerns. However, there is mounting evidence that some shared household toilets have a comparable level of service as private toilets. This study examined the factors that contribute to the quality of shared household toilets in low-income urban communities in Ghana and Kenya. The study design comprised household surveys and field inspections. Overall, 843 respondents were interviewed, and 838 household shared sanitation facilities were inspected. Cleanliness scores were computed from the facility inspections, while a total quality score was calculated based on 13 indicators comprising hygiene, privacy, and accessibility. Regression analyses were conducted to determine predictors of cleanliness and the overall quality of the shared sanitation facilities. More than four out of five (84%) shared toilets in Ghana (N = 404) were clean, while in Kenya (N = 434), nearly a third (32%) were clean. Flush/pour-flush toilets were six times (*p* < 0.01 aOR = 5.64) more likely to be clean. A functional outside door lock on a toilet facility and the presence of live-in landlords led to a threefold increase (*p* < 0.01 aOR = 2.71) and a twofold increase (*p* < 0.01 aOR = 1.92), respectively in the odds of shared sanitation cleanliness. Sanitation facilities shared by at most five households (95% CI: 6–7) were generally clean. High-quality shared toilets had live-in landlords, functional door locks, and were water-dependent. Further studies on innovative approaches to maintaining the quality of these high-quality shared toilets are needed to make them eligible for classification as basic sanitation considering the increasing reliance on the facilities.

## 1. Introduction

In 2019, two billion people were without access to basic sanitation services worldwide [1]. Basic sanitation service refers to an improved toilet facility that is not shared with other households [2]. However, in low-income urban communities, the only alternative to open defecation is shared sanitation—an otherwise improved toilet facility shared between two or more households [3]. Worldwide, the reliance on shared sanitation has nearly doubled since 1990—from 6% to 11% of the global population [4]. Shared sanitation facilities serve about one third of people living in urban communities in sub-Saharan Africa [2]. In Ghana, for example, close to two thirds (60%) of urban residents use shared sanitation, while in Kenya, approximately half of the urban dwellers rely on shared sanitation [2].

However, sanitation researchers and public and environmental health practitioners view shared sanitation differently. To some, shared sanitation facilities are unsafe because they are often unhygienic, do not separate human excreta from human contact, and promote the spread of diarrheal diseases [5,6,7,8]. In a systematic review of 22 studies in 21 countries supporting this notion, Heijnen et al. [4] reported increased odds of diarrheal disease associated with the use of shared sanitation. Other studies worldwide have also found evidence that shared sanitation facilities are poorly managed [9,10]; costly to construct [11,12,13]; do not cater to the needs of women and children [11]; and most users are not satisfied with the hygiene conditions [14,15]. The latest report by the Joint Monitoring Programme (JMP) classifies shared sanitation as a limited sanitation service [2]. This creates the impression that investing in shared sanitation does not count towards achieving universal access to sanitation by 2030, per Target 6.2 of the Sustainable Development Goals. Consequently, donors and governments are unwilling to invest in shared sanitation in low-income communities, thereby worsening the plight of the poor [3].

Conversely, proponents of shared sanitation argue that it is the only viable sanitation service for low-income communities considering the economic, technical, and social conditions [16,17,18,19]. However, although the global target is to achieve universal access to sanitation by 2030, it is unfeasible to provide each household, especially in low-income communities, with a private toilet owing to, among others, overcrowding and poverty [20,21,22]. Particularly, empirical evidence from India and Tanzania shows that toilet facilities shared by neighbouring households in the same compound are more accessible, hygienic, and less contaminated with pathogens compared to communal and some improved household sanitation facilities [23,24].

Although dependence on shared sanitation is increasing globally, there is insufficient evidence to underscore the conditions under which shared sanitation can provide a safe, affordable, and acceptable substitute to private household latrines. It is apparent that if dependence on shared sanitation continues to increase, but is not considered as basic sanitation, universal access to safe sanitation may never be achieved. Some studies [3,17,22] have suggested certain criteria for classifying shared sanitation as basic sanitation. However, there is limited empirical evidence to support this, and the debate on shared sanitation remains open. To the best of our knowledge, only one empirical study [25] has been undertaken to define acceptable or high-quality shared sanitation, by quantitatively assessing the determinants of the overall quality of shared sanitation in low-income urban communities across different countries. In their study, eight toilet quality characteristics were distributed across three quality indicators: hygiene, safety, and privacy were aggregated into a single sanitation quality index (Table 1).

However, the study omits a crucial quality dimension in its computation of sanitation quality index: accessibility, contrasting the WHO guidelines on sanitation and health [26]. A clean toilet that is not accessible is of no use. Therefore, the objective of this study is to improve the existing approach to defining the underlying factors influencing the quality of shared household toilets. This study employs 13 toilet quality characteristics across three sanitation quality indicators: hygiene, privacy, and accessibility to compute the quality scores of household shared sanitation (Table 1) and to determine the underlying factors in low-income urban communities in Ghana and Kenya. The indicators used in this study were developed from the guidelines on sanitation and health recommended by the World Health Organisation [26].

Clearly, more research is needed to support the development of criteria to define high-quality or acceptable shared sanitation facilities for inclusion into the basic sanitation service category. This is particularly important as there is no universal definition of high-quality shared sanitation. Therefore, the findings of this study will contribute to efforts to arrive at a set of indicators for defining acceptable shared sanitation in low-income urban communities. At present, classifying all shared sanitation as unacceptable in relation to the SDG6 makes it a disincentive for donor organisations to invest in these facilities since it does not count towards basic sanitation. Consequently, a well-defined criteria for accepting high-quality shared sanitation as basic sanitation would help attract investment to improve access to sanitation in these communities and inform policy actions towards achieving Sustainable Development Goal 6.

## 2. Materials and Methods

### 2.1. Study Areas

In Kenya, the study was conducted in Kisumu, the third largest city after Nairobi and Mombasa. Kisumu has an estimated population of more than 1 million as of 2019, and 60% of this population lives in low-income settlements characterised by overcrowding, poor housing, and lack of basic services [27,28]. Basic services such as water, sanitation, and solid waste disposal are shared among households. Due to the lack of a sewer system in the settlements, pit latrines are the most common technology. Others such as ecological sanitation and septic tanks are also used. A previous study estimates that approximately half of the compounds in Kisumu’s informal settlements lack sanitation facilities [29]. Sharing of sanitation facilities at the compound level is common. The main slum communities are Nyalenda A, Nyalenda B, Manyatta A, Manyatta B, Bandani, Obunga, Manyatta Arab, Kaloleni, Kibos, and Nyamasaria.

Nyalenda A and B together constitute the largest informal settlement in Kisumu. Houses are mostly made of mud and iron sheets, although in recent times, there has been some improvement in housing structures in more modern housing. Pit latrines are common, and where sanitation facilities are unavailable, residents use open spaces and polythene bags. The study was carried out in Nyalenda A.

In Ghana, the study was conducted in Kumasi, the second largest and fastest-growing city in the country. It is located about 270 km north-west of the national capital, Accra, and covers a total land surface area of a little more than 200 square kilometres [30]. The city has about 2.5 million people with an annual growth rate of nearly 5%. It is completely urbanised and has seen the growth and expansion of slum settlements over the years [31]. These settlements have mainly been indigenous/migrant areas that have expanded to accommodate the city’s low- and medium-income earners and fringe communities that have been incorporated into the city as a result of the city’s expansion [32]. Access to sanitation is a crucial challenge in the city as about 40% of the city’s population relies on public toilets while 2.5% have no toilet facility [33].

The study was conducted in three low-income areas in Kumasi-Ayigya-Zongo, Ahwiam, and Accra town. These towns are located along the main Accra–Kumasi road and are characterised by poor housing, poor access to services, and poor sanitation [32]. Most of the housing in the settlement are single-storey traditional compound houses inhabited by more than one household [34] (Dinye and Acheampong, 2013). Generally, shared sanitation in low-income communities in Kumasi are flush toilets and can be shared by multiple households [35,36].

### 2.2. Study Design, Sample Size, and Sampling

The study was a cross-sectional survey, and to determine the sample size, the following formula was adopted:$$n=\frac{z^{2}\;\times \;p\;\left(1-p\right)}{d^{2}}$$

The standard normal variate (z) at the 95% confidence interval was used (1.96), and the expected proportion of compounds sharing sanitation facilities (p), was 50%. The study assumed a precision level (d) of 5% and a 10% non-response rate. This resulted in a sample size of 384 compounds with shared sanitation facilities in Kenya, which was increased to 422 compounds based on the non-response rate. A similar calculation was done in the case of Ghana, and the expected sample size, including the non-response rate was 427.

To ensure that the compounds were selected appropriately, transect walks were carried out in the settlements to identify boundaries of each unit. These transect walks were conducted by the Community Health Volunteers (CHVs), the Environmental Health Officers (EHOs), and the researchers. Research Assistants worked with CHVs/EHOs to facilitate movement within the settlements. The CHV/EHOs and Research Assistants identified a starting point for each unit and selected the next immediate compound as the first compound. A compound was selected only if it had a shared sanitation facility. In each compound, enumerators randomly determined the household by selecting a random number and identifying the household corresponding to the number by counting from the left. If the household was unavailable, the Research Assistants again randomly selected another number from the pool of random numbers. The next compound was selected by skipping every two compounds. This process continued as the Research Assistants moved towards the end of the unit.

### 2.3. Data Collection and Analysis

For easy data collection and entry, data were collected on tablets using the Open Data Kit (ODK) software. The trained Research Assistants completed the guides, and after each interview, the data were uploaded to a central repository and cross-checked for any errors.

A structured questionnaire was developed and pre-tested in similar but different low-income settlements in both Kenya and Ghana. The questionnaire contained distinct sections to collect data on the socio-demographic profile of latrine users, characteristics of the existing shared sanitation facilities, and management practices. Under socio-demographic characteristics, the questions were focused on the location of the respondent, gender, age, educational level, marital status, occupation, monthly income, and tenure status of the compound house. For characteristics and management of the toilet facilities, the questions were centred on the type of toilet facility, location of the toilet, number of households sharing the facility, cleaning arrangements, cleaning frequency, presence of a landlord, involvement of landlords in cleaning the toilet facilities, and observed cleanliness.

We interviewed as many household heads as possible but when they were unavailable, a representative of the household head was interviewed. Field staff also inspected the shared sanitation facilities after each interview. Research Assistants who supported data collection were trained on various aspects of the study for data quality purposes. All the sanitation facilities observed by the Research Assistants were categorised into four based on the level of cleanliness: very clean, clean, dirty, and very dirty. This was complemented by taking photos of the sanitation facilities for comparison and ensuring that the correct category was assigned to each facility. Results from the pre-testing were used to refine the tool before the actual survey.

Data were cleaned in Microsoft Excel and analysed with R Studio (version 1.3.1093). Descriptive statistics were first performed, and to understand the factors that influence the cleanliness of the observed shared sanitation facilities, the level of cleanliness was regressed against six predictors (independent variables): the presence of a functional door lock outside the facility; type of sanitation facility; cleaning routine; involvement of landlords in cleaning the toilet facility; the presence of live-in landlord; and the number of households sharing the facility. Those facilities classified as either clean or very clean were coded as 1, while the others were coded as 0. Eventually, multiple logistic regression was used to determine which predictors significantly influenced the cleanliness of the toilet facilities.

The total quality of the sanitation facilities was also determined based on three main factors: hygiene; privacy, and accessibility. These factors were selected based on desirable qualities of shared sanitation by users reported in literature [37,38,39,40] and similar studies conducted earlier [41,42]. Under each of these factors, a set of indicators was used to calculate the score for the specific factor (see Appendix A). Overall, 13 indicators were used to compute the total quality score (TQS). The TQS for a sanitation facility was subsequently obtained by summing the scores for all three quality factors. Multiple linear regression was used to determine which predictors significantly influenced the total quality score of the toilet facilities. Again, the same independent variables used for the multiple logistic regression against cleanliness were used.

### 2.4. Ethical Approval

In Kenya, ethical approval was obtained from the institutional ethics committee of GLUK (Ref GREC/001/285/2018), and a research permit from the Kenya National Council of Science and Technology (Ref: NACOSTI/P/18/5546/24979). Similar ethical approval for this study was received from the Centre for Scientific and Industrial Research (CSIR) in Ghana, with Ref: CSIR/IRB/PI/VOL1.

Research Assistants were trained extensively on ethical issues in research and standard operating procedures (SOPs) related to obtaining consent and the rights of respondents. Respondents received a written information sheet detailing their right to complete information on the research, their right to withdraw from the study if they chose to, the risks involved, and the persons they should contact in case of any further questions and/or concerns. Respondents consented by signing a consent form before being interviewed.

## 3. Results and Discussion

The socio-demographic characteristics of the respondents are shown in Table 2. The study involved 843 respondents comprising 410 Ghanaians and 433 Kenyans (one respondent per compound), constituting a response rate of 99%. In the case of Kenya, actual field data collection ended up with 433 respondents (compounds) even though a minimum sample size of 422 was anticipated. In both countries, females constituted about three quarters of the respondents (Table 2). In terms of age, most Kenyan respondents (71%) were in the youthful age bracket (18–35), but among Ghanaian respondents, more than half (57%) were more than 35 years. The mean age of Ghanaian respondents was 41 years (95% CI: 40–43), while on average, Kenyan respondents were 33 years (95% CI: 32–34). More than half of respondents in both countries had at least some secondary education (56% in Ghana; 58% in Kenya), and most were married (56% in Ghana; 87% in Kenya). Two thirds of Ghanaian respondents (66%) were self-employed, while about one third of Kenyans were self-employed, with slightly less than half (42%) of Kenyan respondents being casual workers. Half of the Ghanaian respondents reported monthly incomes of less than USD 100, but about two thirds of Kenyan respondents reportedly earned at least USD 100 per month. While about four out of every five houses in Ghana had a live-in landlord, only a third of compound houses in Kenya had live-in landlords.

More than half (57%) of the shared sanitation facilities inspected in Ghana were flush/pour-flush toilets while in Kenya, about 80% of sanitation facilities inspected were pit latrines with concrete slabs (Table 3). The dominance of flush/pour-flush toilets among the shared sanitation facilities in this study is consistent with findings reported by Foggitt et al. [36] in the Kumasi Metropolis. It is in contrast with other studies which found pit latrines to be the dominant shared sanitation technology [37,45]. In Kenya, the distribution of shared sanitation technologies reflects the findings from earlier studies [42,46]. Almost all (96%) of the sanitation facilities in both countries were within the compound premises, while the rest were within users’ reach though outside the compound. Generally, studies have shown that distant toilet facilities discourage users from using them, and for girls and women especially, this puts them at risk of abuse during the night [47,48,49,50]. Therefore, toilet facilities must be located closer to users to ensure that they can be reached easily and used when needed.

On average, there were fewer households sharing toilet facilities in Ghana compared to Kenya. In Ghana, the average number of households sharing a toilet facility was six (95% CI: 6–7), while an average of eight households (95% CI: 8–9) shared a toilet facility in Kenya. Additionally, there were a higher proportion of households (44%) in the 2–4 household category sharing toilet facilities in Ghana while in Kenya, the numbers of households were spread throughout the household categories (Table 3). The number of households sharing a toilet and its impact on the shared facilities’ cleanliness has been a subject of a global debate among public health and sanitation researchers and practitioners. It is one of the primary reasons why the WHO/UNICEF Joint Monitoring Programme for Water and Sanitation deems them undeserving of being included in the basic sanitation service category [51]. This is based on the conception that there is a lack of co-operation in maintaining a clean toilet when sanitation facilities are shared. In line with this, Guenther et al. [52], in a study involving 1500 randomly selected toilets in Kampala, argued that private toilets are generally cleaner than shared toilets, but recommended that toilets shared by at most four households can be classified as improved. This could imply that the relatively higher number of households sharing toilet facilities in Kenya could influence its cleanliness. However, other empirical studies [24,53,54] argued against this notion, indicating that having fewer households sharing a toilet facility does not usually translate into clean toilets and that the focus must go beyond user numbers.

Cleaning arrangements for the shared sanitation facilities in both countries were similar. Tenants generally were responsible for cleaning the toilets. This finding is consistent with other studies on cleaning arrangements for shared sanitation [37,38]. In more than half (57%) of compound houses in Ghana, all tenants were involved in cleaning the toilets while in less than half (44%) of compounds in Kenya, all tenants were involved in cleaning the toilets (Table 3). Comparing the self-reported cleaning frequencies of the toilet facilities shows that Ghanaian households clean their toilets more frequently than Kenyan households. In Ghana, more than two thirds (72%) (7% in Kenya) of the respondents reported cleaning their shared toilet facility daily while in Kenya, a third (33%) (14% in Ghana) of the households reportedly cleaned weekly and about a quarter (24%) cleaned the toilets twice per week. The cleaning arrangements in both countries could be explained by landlords’ presence in the compound houses, though the presence or absence of a landlord did not have a statistically significant association with the cleanliness of toilets as explained under the regression section. Maintaining hygienically clean toilets is necessary to encourage users to use them and avoid potential health risks. Kwiringira et al. [48], in their study to assess the factors that discourage latrine use, found that lack of cleaning of a latrine is among the key factors that trigger users to switch to open defecation. Based on this, it could be inferred that the infrequent cleaning of the latrines, particularly in Kenya, could influence users to switch to other unsafe defecation behaviours. Households must therefore make arrangements that suit their needs to ensure that the facilities are cleaned regularly. These arrangements can include paying a janitorial service provider to clean the toilet facilities adding toilet cleaning fees to monthly rent charges or utility fees to cater for cleaning, and sanctioning residents who abuse toilet use. Authorities should also adopt a carrot-and-stick approach to ensure that shared toilets are maintained. Per this approach, compound houses that maintained clean toilets can be randomly selected and rewarded while those who consistently have poorly maintained toilets can be sanctioned.

Similar to earlier studies [37,38], this study found that more compound houses have live-in landlords in Ghana than in Kenya and this could partly explain why reported cleaning frequencies of the sanitation facilities are higher than in Kenya. Shared sanitation facilities in Ghana were comparatively cleaner than those in Kenya. While more than half of the toilets in Ghana were rated as either clean or very clean, only a third of shared toilets in Kenya was ranked in this category. Consistent with other studies [42,52], this study found that clean shared toilets are usually shared by fewer (Median = 5; 95% CI: 6–7) households than dirty shared toilets (Median = 7; 95% CI: 8–9).

In total, 838 shared sanitation facilities were inspected. Out of this, slightly more than half (52%; *n* = 433) were in Kenya. Figure 1 shows a comparison between shared sanitation facilities in Ghana (GH) and Kenya (KE) in terms of hygiene, privacy, and access. As has been well established in available literature [15,37,49,53,54], several factors come into play when users are defining an ideal shared sanitation facility. For instance, in Uganda, Kwiringira et al. [49] reported that users pointed to cleanliness, privacy, and easy reach as crucial features of their ideal shared toilet. Certainly, clean shared sanitation facilities that do not offer privacy and are not always accessible cannot pass for an ideal toilet. Therefore, the three indicators used in this study provide a holistic assessment framework for the quality of a shared sanitation facility in line with user preferences.

Generally, the findings reveal that Ghana’s shared sanitation facilities outperformed those in Kenya in all aspects (Figure 1). Regarding hygiene conditions, shared sanitation facilities in Kenya mostly satisfied three out of six pre-determined indicators of a hygienic shared toilet (median hygiene score = 3, 95% CI: 2.54–2.73) (Appendix A) while those in Ghana usually satisfied all six indicators (median hygiene score = 6, 95% CI: 4.99–5.23). The situation in Ghana deviates from existing literature [15,37,42,55,56,57], which report that most shared sanitation facilities are unhygienic. Filthy toilets are a source of diarrheal diseases and usually discourage people from using them [39,56,58]. This is what makes them unacceptable to be considered as basic sanitation service per UNICEF and WHO [51]. However, the finding from this study shows that there is a likelihood for some shared sanitation to perform better compared to private toilets in terms of hygiene. These findings can further be consolidated through field inspections over a period to determine whether the hygiene conditions can be maintained. One-off spot checks of shared sanitation facilities might not provide a more holistic picture than multiple random field inspections of the same facilities undertaken over a period.

In terms of privacy, shared sanitation facilities in Ghana usually met all five indicators while in Kenya, four out of the five indicators of privacy were met (Figure 1). Lack of privacy of a toilet facility, as earlier studies [39,40,59,60] have shown, is among the key factors that catalyse reversion to open defecation. Especially for women, privacy largely influences their choice to use a toilet facility [61]. As Kwiringira et al. [49] puts it, for women, using a toilet is much more than relieving oneself. Therefore, an ideal shared sanitation must, besides being clean, tick the box for privacy and accessibility to cater to the needs of all users.

Performance in terms of access to the shared sanitation facilities was comparable between the two countries. Having unrestricted access to sanitation facilities is critical to discourage open defecation. Contrary to the findings of this study, Foggitt et al. [36] reported that a third of residents in households with toilets were prevented from using the toilet facilities in a suburb of the Kumasi Metropolis. This practice has been noted in literature [39] to promote open defecation.

Aggregating these three factors: hygiene, privacy, and accessibility, into a total quality score provides a more holistic picture of how these sanitation facilities cater to the needs of all users [41,42]. Results from multiple linear regression showed that the total quality of shared sanitation is significantly influenced by the presence of a door lock outside the facility (*p* < 0.05; aOR = 5.5), the presence of a landlord (*p* < 0.05; aOR = 3.1), and the type of sanitation facility (*p* < 0.05; aOR = 7.7). Together, these predictors in the model explain a little more than a third (36%) of the variance in the total quality score of the shared sanitation (R^2^ = 0.36). Conversely, the number of households sharing the facility, the landlords’ involvement in cleaning the facility, and the cleaning routine did not significantly influence the total quality of the shared sanitation facilities.

Per the results of the multiple logistic regression analysis (Table 4), the cleanliness of shared sanitation facilities in Ghana was significantly influenced by the presence of a door lock, the frequency of cleaning, the type of sanitation facility, and the involvement of a landlord in cleaning the toilet facility. Conversely, the presence or absence of a landlord and the number of households sharing the toilet facility did not have a statistically significant association with the cleanliness of toilets.

Shared sanitation facilities with a functional door lock outside were three times more likely to be clean (*p* = 0.01; aOR = 3.29) and water-dependent sanitation facilities were three times more likely to be clean (*p* = 0.002; aOR = 2.56) compared to dry types of toilets. Cleaning shared facilities daily also affected cleanliness positively (*p* = 0.001; β = 1.03). Shared sanitation facilities that were cleaned daily by residents were more likely to be clean. The study found that involving landlords in the cleaning of the toilet facility had a negative influence on the cleanliness of toilets. Compound houses that involved landlords in cleaning were 53% less likely to have a clean toilet (*p* = 0.05; aOR = 0.47).

In Kenya, however, none of the predictors were found to influence the cleanliness of the shared sanitation facilities significantly (Table 4). Similar to the results from Ghana, the presence of a door lock outside the toilet facility and the type of sanitation facility positively influenced the cleanliness of the toilet facilities. Likewise, the presence of a landlord and the number of households sharing the toilet negatively affected the cleanliness of the toilet facilities, albeit not to a statistically significant extent. In contrast to the findings in Ghana, the study found that, in Kenya, involving landlords in cleaning the toilet facility positively influenced the cleanliness.

Overall, the study found that four factors influence the cleanliness of shared toilets: the presence of a functional outside door lock; the type of sanitation facility; daily cleaning of the toilet facility; and the presence of a landlord in the compound. Flush or pour-flush toilets were six times (*p* < 0.05; aOR = 5.64) more likely to be clean than dry toilets while those with a functional outside door lock were three times (*p* < 0.05; aOR = 2.71) more likely to be clean (Table 4). Moreover, compound houses with live-in landlords are twice (*p* < 0.05; aOR = 1.92) as likely to have clean toilets and a daily cleaning routine also increases the odds of having a clean toilet by about 40% (*p* = 0.05; aOR = 1.39).

On the contrary, the number of households sharing the toilet facilities and landlords’ involvement in cleaning shared toilet facilities did not influence cleanliness. Although the number of households (increasing number of households) sharing the latrine was found to affect the cleanliness negatively, it was not statistically significant (β = −0.03; *p* = 0.08). Generally, most research focusing on the factors that affect the cleanliness of shared sanitation in low-income communities have concluded that the fewer the number of households sharing a toilet facility, the cleaner the facility [6,62,63]. However, only a few studies have been able to pinpoint the minimum acceptable number of households that can share a toilet facility. Although the UN Joint Monitoring Programme for Water and Sanitation stipulates that toilets shared by at least two households are unhygienic and therefore not considered as basic sanitation, some researchers argue otherwise. For instance, a growing body of literature [51,52,64] has contended that toilets shared by 3–5 households can be classified as improved sanitation.

These facilities used by a limited number of households usually have been shown to have better hygienic conditions than public shared sanitation facilities [22,24]. Clearly, it is crucial to establish some criteria for classifying household shared sanitation as basic sanitation service instead of lumping them together as shared sanitation facilities. As this study has shown, the number of users should not be the only basis for classifying household shared sanitation as other studies have argued. The cleanliness of toilets is also affected by the presence of a functional outside door lock; the type of sanitation facility; daily cleaning of the toilet facility; and the presence of a landlord in the compound. Establishing the cleanliness of shared toilets should be based on multiple spot checks over a period to provide a realistic picture of the likelihood of the cleanliness being maintained—“improved” shared toilets must be kept clean at all times. Since the hygiene condition of a toilet facility is such an essential characteristic of a toilet facility, more focus should be on measures that can be instituted to ensure that shared toilets are always clean. Local authorities can provide leadership in this regard by instituting a carrot-and-stick approach to maintain cleanliness of toilets—rewarding compound houses and constructively addressing those who violate the rules.

Further studies are needed to understand how this approach can be implemented in different cultural settings. Apart from cleanliness, the criteria for “improved” shared sanitation must include privacy and accessibility indicators since these have been reported in various studies as desirable characteristics of an ideal toilet [23,40]. However, extensive multi-country studies are required to further understand the desirable characteristics of an “improved” shared sanitation from different users’ perspective from different cultural settings to inform policy actions.

## 4. Practical Implications of This Study

Based on the insights obtained from the study, it has become evident that some household shared sanitation can potentially be classified as basic sanitation. Apart from cleanliness, shared sanitation can offer privacy and be accessible. On the back of extensive multi-country studies, the WHO/JMP needs to develop a clear definition for acceptable household shared sanitation to attract investment into the low-income urban sanitation sector. This will ensure that local authorities and decision makers adopt measures to improve existing and new toilet facilities for low-income urban dwellers. Such measures could include targeted subsidies and reward schemes to the urban poor communities that are able and willing to provide acceptable shared sanitation worthy of being classified as basic sanitation.

## 5. Conclusions

Defining and classifying some household shared sanitation as basic sanitation service would significantly impact progress towards Sustainable Development Goal 6. It will attract investments into low-income communities to progressively improve the sanitation situation in these communities. The study findings provide valuable insights into the quality of shared toilets and the factors affecting their cleanliness in line with establishing some minimum criteria for improved shared sanitation. Clean toilets protect the health of users and keep users from practising open defecation. This study found that more than half of shared toilets in both countries were clean. Shared toilets are more likely to be clean when they are water-dependent (flush/pour-flush); have a functional outside door lock; are in compounds with live-in landlords; and cleaned daily by residents. The overall quality of shared toilets was significantly associated with all these factors except cleaning routine. While this study found no statistically significant association between a toilets’ cleanliness and the number of households sharing it, overall, clean toilets had relatively fewer households than dirty toilets. Future studies should employ multiple spot checks to establish whether the cleanliness of shared toilets can be maintained over a long period and that they are not one-off. Innovative ways of encouraging shared toilet users to maintain their cleanliness also need to be further studied. A carrot-and-stick approach to reward and sanction users and the engagement of janitorial businesses to clean shared toilets must be tested. Multi-country studies are required to further understand the desirable characteristics of an “improved” shared sanitation from users’ perspectives to inform the development of evidence-based criteria for defining “improved” shared sanitation.

## Figures and Tables

**Figure 1 ijerph-19-04271-f001:**
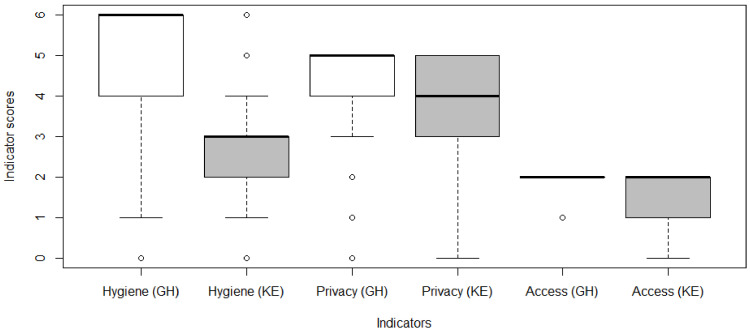
Comparison of indicators of sanitation quality for Ghana and Kenya.

**Table 1 ijerph-19-04271-t001:** Comparison of indicators used in previous studies and the current study.

Existing Literature [25]		This Study	
Quality Dimensions	Indicators	Quality Dimensions	Indicators
Hygiene	1. Solid waste inside the cubicle	Hygiene	Visible faecal matter on the slab/seat
	2. Visible faeces in or around the manhole/pan		Flies on the facility
	3. Insects inside the cubicle		Noticeable odour on the facility
	4. Handwashing facility with soap		Visible urine/saliva on the facility
	5. Clogged in the case of a flush toilet or full in the case of a pit latrine		Maggots in the toilet cubicle
Safety	6. Solid roof (without holes)		Rodents on the facility
	7. Solid floor (without cracks/holes)	Privacy	Presence of a door
Privacy	8. Solid wall		Presence of a door locking latch inside the cubicle
			Presence of a door locking latch outside the cubicle
			Door offers privacy
			Superstructure offers privacy
		Accessibility	Everyone in the house uses the toilet facility
			Toilet facility accessible at all times

**Table 2 ijerph-19-04271-t002:** Socio-demographic characteristics of study respondents.

Characteristics	Ghana		Kenya		Total *n* (%)
	*n*	%	N	%	
**Gender**					
Female	292	71.2	339	78.3	631 (74.9)
Male	118	28.8	94	21.7	212 (25.1)
**Age groups**					
18–25	64	16.2	141	32.6	205 (24.8)
26–35	105	26.6	163	37.6	268 (32.4)
36–45	91	23.0	67	15.5	158 (19.1)
46–55	68	17.2	31	7.2	99 (12)
More than 55 years	67	17.0	31	7.2	98 (11.8)
**Educational level**					
None	54	13.4	9	2.1	63 (7.5)
Primary (not completed)	51	12.6	66	15.2	117 (14)
Primary (completed)	74	18.3	105	24.2	179 (21.4)
Secondary (not completed)	58	14.4	86	19.9	144 (17.2)
Secondary (completed)	104	25.7	108	24.9	212 (25.3)
Tertiary	63	15.6	59	13.6	122 (14.6)
**Marital status**					
Divorced/Separated	21	5.2	4	0.9	25 (3)
Married	226	55.8	376	86.8	602 (71.8)
Single	126	31.1	49	11.3	175 (20.9)
Widowed	32	7.9	4	0.9	36 (4.3)
**Occupation**					
Casual worker	17	4.5	173	42.1	190 (24)
Formal employment	32	8.4	32	7.8	64 (8.1)
Unemployed	81	21.3	70	17.0	151 (19.1)
Self-employed	250	65.8	136	33.1	386 (48.8)
**Monthly income (USD)** *					
<100	149	49.8	139	37.5	288 (43)
100–200	90	30.1	138	37.2	228 (34)
201–300	31	10.4	51	13.7	82 (12.2)
>300	29	9.7	43	11.6	72 (10.7)
**Tenure status of compound house**					
Tenants and landlord	314	79.3	157	36.3	471 (56.8)
Tenants only	60	15.2	252	58.2	312 (37.6)
Tenants with caretaker	22	5.6	24	5.5	46 (5.5)

* Exchange rate for October, 2019: 1 USD = GHS 5 [43]; 1 USD = KES 109 [44].

**Table 3 ijerph-19-04271-t003:** Characteristics and management of shared toilet facilities in Ghana and Kenya.

Characteristics	Ghana		Kenya		Total *n* (%)
	Frequency (*n*)	Percentage (%)	Frequency (*n*)	Percentage (%)	
**Type of toilet**					
Composting toilet	1	0.2	1	0.2	2 (0.2)
Container-based toilet	4	1.0			4 (0.5)
Ventilated improved pit	96	23.8	60	13.9	156 (18.6)
Pour-flush toilet to pit latrine	12	3.0	24	5.5	36 (4.3)
Pour-flush toilet to septic tank	17	4.2	3	0.7	20 (2.4)
Traditional pit latrine with concrete slab	73	18.1	345	79.7	418 (49.9)
Water closet toilet to septic (flush toilet)	201	49.8			201 (24)
**Location of toilet**					
Outside compound	5	1.2	25	6.0	30 (3.6)
Within compound	399	98.8	395	94.0	794 (96.4)
**Number of households sharing the facility**					
2–4	168	44.1	116	26.8	284 (34.9)
5–7	105	27.6	118	27.3	223 (27.4)
8–10	50	13.1	91	21.0	141 (17.3)
More than 10	58	15.2	108	24.9	166 (20.4)
**Cleaning arrangement**					
All tenants	213	57.0	180	44.3	393 (50.4)
All tenants and landlord	66	17.6	47	11.6	113 (14.5)
Specific tenants	54	14.4	36	8.9	90 (11.5)
Specific tenants and landlord	8	2.1	-	-	8 (1)
Women only	16	4.3	-	-	16 (2.1)
Someone paid to clean the facility	2	0.5	-	-	2 (0.3)
Anyone who volunteers to clean	15	4.0	143	35.2	158 (20.3)
**Cleaning frequency**					
Daily	274	71.9	26	7.1	300 (40.3)
Every other day	31	8.1	80	22.0	111 (14.9)
Twice per week	22	5.8	87	23.9	109 (14.6)
Weekly	54	14.2	121	33.2	175 (23.5)
Every other week			10	2.7	10 (1.3)
Monthly			40	11.0	40 (5.4)
**Observed cleanliness**					
Very clean	119	29.5	15	3.5	134 (16.0)
Clean	220	54.5	125	28.8	345 (41.2)
Dirty	64	15.8	210	48.4	274 (32.7)
Very dirty	1	0.2	84	19.4	85 (10.1)

**Table 4 ijerph-19-04271-t004:** Predictors of the cleanliness of shared sanitation facilities.

	Ghana				Kenya				Overall			
	β	aOR	SE	*p* (95% CI)	β	aOR	SE	*p* (95% CI)	β	aOR	SE	*p* (95% CI)
Presence of functional outside door lock	1.11	3.03	0.45	0.01 (0.19–1.98) *	0.37	1.45	0.29	0.20 (−0.18–0.96)	1.00	2.71	0.23	<0.05 (0.56–1.45) *
Type of sanitation facility	0.94	2.56	0.30	0.002 (0.35–1.55) *	0.37	1.45	0.45	0.42 (−0.55–1.25)	1.73	5.64	0.20	<0.05 (1.35–2.14) *
Cleaning toilet daily	1.03	2.80	0.31	0.001 (0.40–1.68) *	−0.24	1.27	0.23	0.30 (−0.69–0.21)	0.33	1.39	0.17	0.05 (−0.002–0.66) *
Involving landlords in cleaning toilet facility	−0.75	0.47	0.40	0.05 (−1.45–0.01) *	0.70	0.5	0.39	0.08 (−0.08–1.47)	−0.0007	1.00	0.26	1.00 (−0.51–0.51)
Presence of landlord	−0.05	0.95	0.40	0.91 (−0.86–0.71)	−0.51	0.60	0.28	0.07 (−1.09–0.03)	0.65	1.92	0.16	<0.05 (0.33–0.96) *
Number of households sharing toilet facility	−0.02	0.98	0.03	0.49 (−0.09–0.05)	−0.02	0.98	0.02	0.30 (−0.07–0.02)	−0.03	0.97	0.02	0.08 (−0.06–0.003)

* Statistically significant predictor of cleanliness of shared sanitation.

## Data Availability

The data presented in this study are available on request from the corresponding author. The data are not publicly available due to privacy and ethical reasons.

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
