# Peer review of "Shared but Clean Household Toilets: What Makes This Possible? Evidence from Ghana and Kenya"

_ijerph, 2022, doi:10.3390/ijerph19074271_

Round 1
Reviewer 1 Report
The paper deals with a an interesting issue of public health and its findings can inform the policies on funding in this important area if nothing else. The methods used seem to be robust but the lack of information about the data collected using the constructed questionnaire (apart from the socio-demographic data) makes it difficult for the reader sometimes to make sense of the data referred to in the discussion. A brief statement (in the Materials and Methods) about the data collected via the questionnaire would be useful and adds more clarity.
Author Response
Dear Sir,
Kindly find attached the response to the reviewers
Regards

Reviewer 2 Report
This paper presents statistical results on the actual condition of shared toilets in Ghana and Kenya. Although the topic is written interesting and easy to read, I think there is room for improvement as follows:
- The description of related work needs to be improved. The authors refer to 65 papers, but they do not seem to have summarized them clearly in this paper.
- As a result of the analysis, the authors show that the status of the shared toilets in Kenya is better than in Ghana. Are there any reasons or facts to support this?
Author Response
Dear Sir,
Kindly find attached response to the reviewers.
Regards

Reviewer 3 Report
This article aims to examine the determinants of the quality of shared household toilets in low-income urban communities in Ghana and Kenya.
The article has a good approach and a logical sequence that is favorable for reading, but some points still deserve attention:
1- The objective is not clear in the abstract and introduction of the article.
2- The contributions expected from the research should be better explained.
3- In the introductory part, it would be interesting to include a table showing which indicators previous studies used and compare with the indicators in this article.
4- How were the indicators used defined?
5- Based on the discussion of the results, it would be interesting to include a subsection of practical implications, based on the insights obtained.
6- The table of supplementary material could be included in the body of the article.
Author Response
Dear Sir,
Kindly find attached responses to the reviewers
Regards

Round 2
Reviewer 2 Report
Most of my previous concerns have been resolved.
Reviewer 3 Report
The authors satisfactorily responded to the suggestions of this reviewer.